

# Genetic influences on creativity: an exploration of convergent and divergent thinking

Wei Han[1],[*], Mi Zhang[1],[*], Xue Feng[2], Guihua Gong[3],
Kaiping Peng[1] and Dan Zhang[1]

[1] Department of Psychology, Tsinghua University, Beijing, China
[2] Education College, Yangtze University, Jingzhou, Hubei, China
[3] Repconex, Inc, Beijing, China
[*] These authors contributed equally to this work.

## ABSTRACT

Previous studies on the genetic basis of creativity have mainly focused on the biological mechanisms of divergent thinking, possibly limiting the exploration of possible candidate genes. Taking a cognition-based perspective, the present study investigated the genetic basis for both the divergent and the convergent thinking components of creativity. A total of 321 Chinese university students were recruited to complete the Guildford Unusual Using Test (UUT) for divergent thinking capability and the Remote Associates Test (RAT) for convergent thinking capability. The polymorphism of rs2576037 in KATNAL2 was related to the fluency and originality component scores of UUT, and the polymorphism of rs5993883 in COMT, rs362584 in SNAP25 was related to the RAT performance. These effects remained significant after considering the influence of age, gender and intelligence. Our results provide new evidence for the genetic basis of creativity and reveal the important role of gene polymorphisms in divergent and convergent thinking.

## INTRODUCTION

Creativity, the ability to develop new and useful ideas, is the key driving force behind scientific, technological and cultural innovation (*Sternberg & Lubart, 1999*; *Diedrich et al., 2015*). Given the widely acknowledged importance of creativity, studying its psychological basis has attracted much attention in past decades. With recent advances in molecular genetics, exploring the genetic basis of creativity has attracted increasing interest in the field of psychology as well. Finding creativity characteristic genes would help explain the individual differences in creative behaviors, providing a deeper understanding of the biological and the psychological basis of creativity.

The majority of genetic exploration on creativity has taken a biological perspective, with a special focus on the dopamine (DA) system (*Reuter et al., 2006*; *Runco et al., 2011*; *Mayseless et al., 2013*; *Zhang, Zhang & Zhang, 2014a*). Within the DA system, the DA D2 receptor gene (DRD2) and the catechol-O-methyltransferase gene (COMT) are of most interest. Several single nucleotide polymorphisms (SNPs) in DRD2 have been

Corresponding author
Dan Zhang,
dzhang@tsinghua.edu.cn

found to be related to divergent thinking capability measured either by behavioral tests or by self-report (*Reuter et al., 2006*; *Zhang, Zhang & Zhang, 2014a*; *Takeuchi et al., 2015*; *Yu, Zhang & Zhang, 2017*). Among them, the effect of rs1800497 was first reported in the Caucasian population (*Reuter et al., 2006*), with two follow-up reports in Han Chinese population as well (*Yu, Zhang & Zhang, 2017*; *Zhang, Zhang & Zhang, 2014a*; but see *Takeuchi et al., 2015* in which a null result was reported with the Japanese population). Likewise, the polymorphism of rs4648319 was found related to fluency and flexibility components of the Runco Ideational Behavior Scale (*Yu, Zhang & Zhang, 2017*), as well as verbal fluency measured by selected divergent tasks of the Runco Creativity Assessment Battery (rCAB) (*Zhang, Zhang & Zhang, 2014a*). Findings on COMT gene were more controversial. While several SNPs in COMT (rs174697, rs737865, rs5993883, rs4680) were associated with performances in divergent thinking tasks of rCAB (*Zhang, Zhang & Zhang, 2014b*), other studies failed to find any main effect on either divergent thinking performance in the abbreviated torrance test for adults or real-life creative achievement measured by the Creative Achievement Questionnaire (CAQ) (*Zabelina et al., 2016*). Also, individuals carrying the DRD4-7R allele were found to score significantly lower on divergent thinking tasks compared to non-carriers (*Mayseless et al., 2013*). Researchers have also reported creativity characteristic genes other than the DA system. Several SNPs on 5-HT–related tryptophan hydroxylase gene have been found to be associated with the performances in divergent thinking tasks as well (*Reuter et al., 2006*; *Zhang & Zhang, 2017*). One SNP in the oxytocin receptor gene, rs1042778, predicted participants' fluency, flexibility, and originality of divergent creative ideation measured by the Alternative Using Task (*De Dreu et al., 2014*).

Notably, the above-mentioned genetic studies have primarily employed divergent thinking task performances as the behavioral indicator for creativity. While divergent thinking capability is probably the most well studied component of creativity, convergent thinking has received increasing attention in recent years (*Brophy, 2001*; *Lee & Therriault, 2013*). Divergent and convergent thinking processes are believed to provide complementary roles for constituting creative behaviors (*Guilford, 1967*): Whereas divergent thinking refers to the generation of multiple ideas or solutions for a single problem, convergent thinking is associated with finding a single solution to a problem in an analytical and deductive way. The major distinction is that the former starts from a single point and generates multiple ideas, whereas the latter starts from multiple points and seeks for a single solution (*Brophy, 2001*). Accordingly, distinct tasks have been designed to measure these two processes. Divergent thinking tests, such as Guildford Unusual Using Test (UUT) (*Guilford, 1967*), require participants to generate a broad range of ideas to a given stimuli. Remote Associates Test (RAT), on the other hand, has a structure dictated by the definition of convergent thinking, which involves finding a criteria-meeting mediating link to combine stimulus from mutually remote associative clusters (*Mednick, 1962*).

To the best of our knowledge, there are only two recent genetic studies on insight problem solving that are related to convergent thinking. In both studies, classical insight problems were presented, and the performance scores were found to be correlated with

polymorphisms of two SNPs in DRD2 (rs1800467 & rs6278) (*Zhang & Zhang, 2016*) and one SNP in COMT (rs5993883) (*Jiang, Shang & Su, 2015*). As insight problem solving is suggested to reflect a kind of convergent thinking (*Jiang, Shang & Su, 2015*), these results provide preliminary evidence for the possible genetic involvement for convergent thinking.

Creativity can be defined in a broader sense in the cognitive domain beyond the divergent and convergent thinking processes. A variety of basic cognitive functions and capabilities have been reported to be correlated with creativity measures. To create something new, one should first be able to store and retrieve information so that one can easily make associations between concepts or objects (*Avitia & Kaufman, 2014*). Hereby, the creative process can benefit from working memory by switching between response categories and breaking away from ineffective approaches to a solution (*Lee & Therriault, 2013*). Evidence showed that when increasing working memory load, insight problem-solving performance was hindered whereas divergent thinking performance was enhanced (*Lin & Lien, 2013*). Personality traits can influence creativity achievement as well. It has been shown that openness and extraversion contributed to the quantity and diversity of creative products (*Kandler et al., 2016*). Creativity is also linked to general intelligence. Investment theory argues that intelligence is a necessary but not sufficient condition of creativity (*Sternberg, 2012*). Studies showed that intelligence contributed to both divergent thinking and convergent thinking (*Batey, Furnham & Safiullina, 2010*; *Lee & Therriault, 2013*).

Accordingly, studies are suggesting a possible genetic basis for these cognitive functions and capabilities as well. For long-term memory, rs6265 in the BDNF gene has been found associated significantly with performance on the auditory delayed element of the Wechsler Memory Scale (WMS-III) (*Yeh et al., 2012*). The same SNP appeared to be related to visual and spatial components of memory (*Yogeetha et al., 2013*). Also, another SNP, rs17070145, has an association with memory functions (*Witte et al., 2016*). An SNP within COMT gene, rs4680, plays an important role in working memory since there are findings on its association with prefrontal activation during various working memory tasks (*Meyer-Lindenberg et al., 2006*; *Papaleo et al., 2014*). This SNP also has an impact on children's prefrontal cognitive functions (*Diamond et al., 2004*). The genetic basis of personality has also been well documented. Systematically investigations using genome-wide association studies (GWAS) and meta-analysis can be found in literatures, with different results for different nationalities (*Terracciano et al., 2010*; *Kim et al., 2015*; *De Moor et al., 2015*). Notably, rs16921695 and rs912765 were found to be related to openness and extraversion in Asian population (*Kim et al., 2015*). When it comes to intelligence, there are several linkages between SNPs and intelligence supported. The gene locations above, rs4680 and rs6265, are two examples (*Harris et al., 2006*; *Barnett et al., 2007*). Besides, rs42352 has been found to be associated with hippocampal volume in Han Chinese population and further moderate the association between hippocampal volume and Raven's Progressive Matrices performance (*Zhu et al., 2014*).

To sum up, the state-of-the-art investigations on genetic influences of creativity has mainly employed tests for divergent thinking and focused on genes within the DA system.

**Table 1 Descriptive statistics of demographic characteristics.**

|  | N(female)/N(total) | Mean age | SD(age) | Min(age) | Max(age) |
|---|---|---|---|---|---|
| THU | 55/144 | 19.75 | 1.213 | 18 | 23 |
| CAU | 89/142 | 20.35 | 1.284 | 17 | 24 |
| All | 144/286 | 20.05 | 1.283 | 17 | 24 |

**Notes:**

N refers to the number of participants. THU is the abbreviation of Tsinghua University. CAU is the abbreviation of China Agricultural University.

Considering the multifaceted definition of creativity, it is necessary to explore convergent thinking capability as well as other related cognitive functions and capabilities.

In the present study, we explored possible genetic influences of both divergent and convergent thinking capabilities within the same group of participants. In contrast to previous studies that selected SNPs on the basis of hypothesized biological mechanisms, here we selected the SNPs from a cognition-based criterion, taking into consideration the SNPs with a potential contribution to creativity-related cognitive functions and capabilities. Our results are expected to provide complementary evidence toward a complete overview of the genetic influences on creativity.

## METHODS

### Participants

A total of 321 healthy Chinese undergraduates (160 females, mean age 20 years, ranging from 17 to 24 years) recruited from two universities (161 from Tsinghua University and 160 from China Agricultural University) in Beijing participated in the study. The participants were given either course credits (for the participants from Tsinghua University) or cash (40 yuan, for the other participants) for their participation. In addition, individualized gene reports were delivered to all participants free of charge. All participants provided their written consent. The study was conducted in accordance with the Declaration of Helsinki and approved by the local Ethics Committee of Tsinghua University.

A total number of 35 participants were excluded from further analysis: five did not complete the web-based scale; 27 participants failed to pass the attention check item in the web-based scale; three failed to complete the paper-and-pencil task. Therefore, a final number of 286 participants (144 females, 144 from Tsinghua University and 142 from China Agricultural University) was included in the current analysis. The demographic characteristics of the final sample were summarized in Table 1.

### Procedure

To assess the participants' divergent and convergent thinking capability, they first completed a paper-and-pencil measurement composed of the Guildford UUT (*Guilford, 1967*) and the RAT (*Mednick, 1968*; Chinese version from *Xiao, Yao & Qiu, 2016*). Here UUT is used to reflect divergent thinking capability whereas RAT is employed to reflect convergent thinking capability. Both UUT and RAT have time limits: 5 min for each UUT task and 5 min for RAT task. UUT output three component

**Table 2 Related cognitive functions or capabilities of selected SNPs.**

| SNP | Gene | Reference | Related cognitive functions or capabilities |
|---|---|---|---|
| rs1042778 | OXTR (oxytocin receptor) | De Dreu et al. (2014) | Creative ideation in UUT |
| rs1799913 | TPH1 (tryptophan hydroxylase 1) | Reuter et al. (2006) | Figural and numeric creativity |
| rs1800497 | DRD2 (dopamine receptor D2) | | Verbal creativity |
| rs5993883 | COMT (catechol-O-methyltransferase) | Jiang, Shang & Su (2015) | Insight task performance |
| rs42352 | SEMA5A (semaphorin 5A) | Zhu et al. (2014) | Moderation between hippocampal volume and Raven's Progressive Matrices performance |
| rs17070145 | WWC1 (WW and C2 domain containing 1) | Witte et al. (2016) | Memory performance by the Rey Auditory Verbal Learning Test (AVLT) |
| rs4680 | COMT | Meyer-Lindenberg et al. (2006) | Working memory |
| rs6265 | BDNF (brain-derived neurotrophic factor) | Yeh et al. (2012) | Intelligence by the Wechsler Memory Scale (WMS-III) |
| rs1079597 | DRD2 | Fischer, Lee & Verzijden (2018) | Extraversion and neuroticism |
| rs6832769 | CLOCK (clock circadian regulator) | Terracciano et al. (2010) | Agreeableness |
| rs644148 | ZNF180 (zinc finger protein 180) | | Openness |
| rs10251794 | CNTNAP2 (contactin associated protein-like 2) | | Openness |
| rs362584 | SNAP25 (synaptosome-associated protein 25) | | Neuroticism |
| rs12601685 | OR1A2 (olfactory receptor family 1 subfamily A member 2) | Kim et al. (2013) | Neuroticism |
| rs35855737 | MAGI1 (membrane associated guanylate kinase, WW and PDZ domain containing 1) | De Moor et al. (2015) | Neuroticism |
| rs16921695 | IMPAD1 (inositol monophosphatase domain containing 1) | Kim et al. (2015) | Openness |
| rs912765 | LMO4 (LIM domain only 4) | | Conscientiousness |
| rs2576037 | KATNAL2 (katanin catalytic subunit A1 like 2) | De Moor et al. (2012) | Conscientiousness |

scores (fluency, flexibility, and originality) on the participants' divergent thinking capability and RAT reported a single score on the participants' convergent thinking capability. The participants then provided their buccal swabs for genotyping. They also took part in the Raven's Advanced Progressive Matrices (RAPM) test for measuring intelligence. They were asked to fill out a web-based scale later using their own computers for the necessary demographical information (age, gender, student ID, etc.), as well as the CAQ. Detailed explanations on these tests and their scoring methods are provided in the Appendix.

## Genotyping

Based on previous studies as reviewed in Introduction, 18 SNPs (rs10251794, rs1042778, rs1079597, rs12601685, rs16921695, rs17070145, rs1799913, rs1800497, rs2576037, rs35855737, rs362584, rs42352, rs4680, rs5993883, rs6265, rs644148, rs6832769, rs912765) as well as the DRD4 repeat-number polymorphism were chosen in the HapMap database and NCBI SNP database. An overview of all the SNPs included in the present study as well as our considerations are provided in Table 2.

The buccal swab was obtained from each participant, and the DNA was extracted using TIANamp Genomic DNA Kit (Tiangen, Beijing, China). Purified genomic DNA was

eluted in 50 μL elution buffer, and 50 ng of DNA was used for PCR. And the Primer3 software program was used to design primers. The SNP genotyping work was typed by SNaPshot Multiplex kit. The current method for DRD4 repeat-number polymorphism is genotyping based on short tandem repeat (STR)–PCR involving coamplification of a panel of STR loci by multiplex PCR and downstream fragment length analysis, usually performed by capillary electrophoresis.

### Data analysis

We first tested whether the genotype distributions were in Hardy–Weinberg equilibrium and calculated the linkage disequilibrium (LD) patterns of the SNPs with Shesis before further analysis. The genotypes of SNPs were coded dichotomously through a dominant model, for example, combining the homozygotes for less proportion with heterozygotes to balance the sample volume between groups (*Zhang, Zhang & Zhang, 2014a*; *Jiang, Shang & Su, 2015*). The genetic influence on divergent thinking was investigated using multivariate analysis of variance (MANOVA), with the three component scores from UUT as dependent variables. Three post-hoc analyses of variance (ANOVA) analyses were then performed to identify the influential gene loci for each component. The corresponding analyses of covariance (ANCOVA) were conducted in parallel with age, gender, university source (dichotomously coded for the two universities) and RAPM score as covariates in order to control for intelligence and other confounding factors. Prior to performing MANOVA, Box's M tests were conducted to examine the homogeneity of covariance matrices. Likewise, ANOVA and ANCOVA were performed for convergent thinking with the RAT score as dependent variable. The participants' real-life creative achievements and personality traits were not included as they provided information beyond the scope of the present study. These data will be used elsewhere in future studies.

## RESULTS

The mean score of all 286 participants on RAT (reflecting convergent thinking) was 9.56 with a standard deviation (SD) of 2.57. The mean scores of fluency, flexibility, and originality components on UUT (reflecting divergent thinking) were standardized scores with zero mean value. The fluency score ranged from −4.59 to 6.24 (SD = 1.85), the flexibility score ranged from −4.83 to 4.42 (SD = 1.65), and the originality score ranged from −3.30 to 6.83 (SD = 1.80). The pairwise Pearson correlations between the RAT score and the three UUT component scores are shown in Table 3. Significant correlations were observed within the three UUT component scores, but all of them did not significantly correlate with the RAT score. The adjusted CAQ score was significantly correlated with all the three UUT component scores but not the RAT score. The correlation between the creativity measures, age and intelligence are also provided in Table 3.

Table 4 summarizes the minor allele frequency and other detailed information for the 17 SNPs, except for rs35855737 and DRD4 due to their limited diversity in the current sample (TT genotype and Non-7R for all participants at rs35855737 and DRD4, respectively). The genotype distributions of 16 out of 17 SNPs were in Hardy–Weinberg

**Table 3 Pearson correlations of the creativity tasks, age, and intelligence.**

| *p*-value | Flexibility | Originality | RAT | CAQ | Age | Intelligence |
|---|---|---|---|---|---|---|
| Fluency | 0.63** | 0.90** | 0.02 | 0.23** | −0.14* | 0.18** |
| Flexibility | | 0.47** | 0.02 | 0.14* | −0.16** | 0.11 |
| Originality | | | −0.01 | 0.20** | −0.15* | 0.18** |
| RAT | | | | −0.01 | −0.19* | 0.19** |
| CAQ | | | | | −0.11 | −0.01 |
| Age | | | | | | −0.07 |

Notes:
* *p* < 0.05.
** *p* < 0.01.

**Table 4 SNPs genotype frequence.**

| SNP | MAF | Genotype | Frequency | HWE *p* |
|---|---|---|---|---|
| rs10251794 | 0.24 | TT/TA/AA | 0.62/0.34/0.04 | 0.99 |
| rs1042778 | 0.41 | GG/TG/TT | 0.85/0.15/0.01 | 0.88 |
| rs1079597 | 0.25 | CC/CT/TT | 0.34/0.49/0.17 | 0.97 |
| rs12601685 | 0.13 | CC/CG/GG | 0.83/0.17/0.01 | 0.97 |
| rs16921695 | 0.32 | GG/GT/TT | 0.02/0.29/0.68 | 0.84 |
| rs17070145 | 0.48 | CC/CT/TT | 0.06/0.33/0.61 | 0.77 |
| rs1799913 | 0.32 | GG/GT/TT | 0.32/0.45/0.24 | 0.20 |
| rs1800497 | 0.33 | GG/GA/AA | 0.36/0.49/0.14 | 0.78 |
| rs2576037 | 0.44 | CC/CT/TT | 0.21/0.51/0.28 | 0.98 |
| rs362584 | 0.2 | GG/GA/AA | 0.57/0.39/0.04 | 0.34 |
| rs42352 | 0.47 | TT/TA/AA | 0.24/0.46/0.3 | 0.46 |
| rs4680 | 0.37 | GG/GA/AA | 0.59/0.36/0.06 | 0.99 |
| rs5993883 | 0.48 | GG/GT/TG/TT | 0.19/0.43/0.07/0.31 | 0.94 |
| rs6265 | 0.2 | CC/CT/TT | 0.24/0.51/0.24 | 0.89 |
| rs644148 | 0.26 | GG/GT | 0.73/0.26 | 0.03 |
| rs6832769 | 0.28 | GG/GA/AA | 0.08/0.42/0.49 | 0.96 |
| rs912765 | 0.4 | CC/CT/TT | 0.44/0.43/0.13 | 0.67 |

Note:
MAF, minor allele frequency; HWE, Hardy–Weinberg equilibrium.

equilibrium. The LD patterns of the SNPs above were calculated with Shesis (Fig. 1). Strong LD was observed between rs1079597 and rs1800497 ($r^2 = 0.89$). Therefore, rs644148, rs35855737, and DRD4 are excluded in the following analysis for either violation of Hardy–Weinberg equilibrium or limited diversity.

## Genetic influences on divergent thinking

A MANOVA analysis was conducted using fluency, flexibility, and originality component scores from UUT as dependent variables to identify the genes correlated with divergent thinking capability. Box's M tests reported that the data for the comparison groups were of homogeneous covariance matrices for all measures ($p$s > 0.001). Among all SNPs, significant differences were observed between rs2576037 groups ($F(3, 277) = 4.043$,

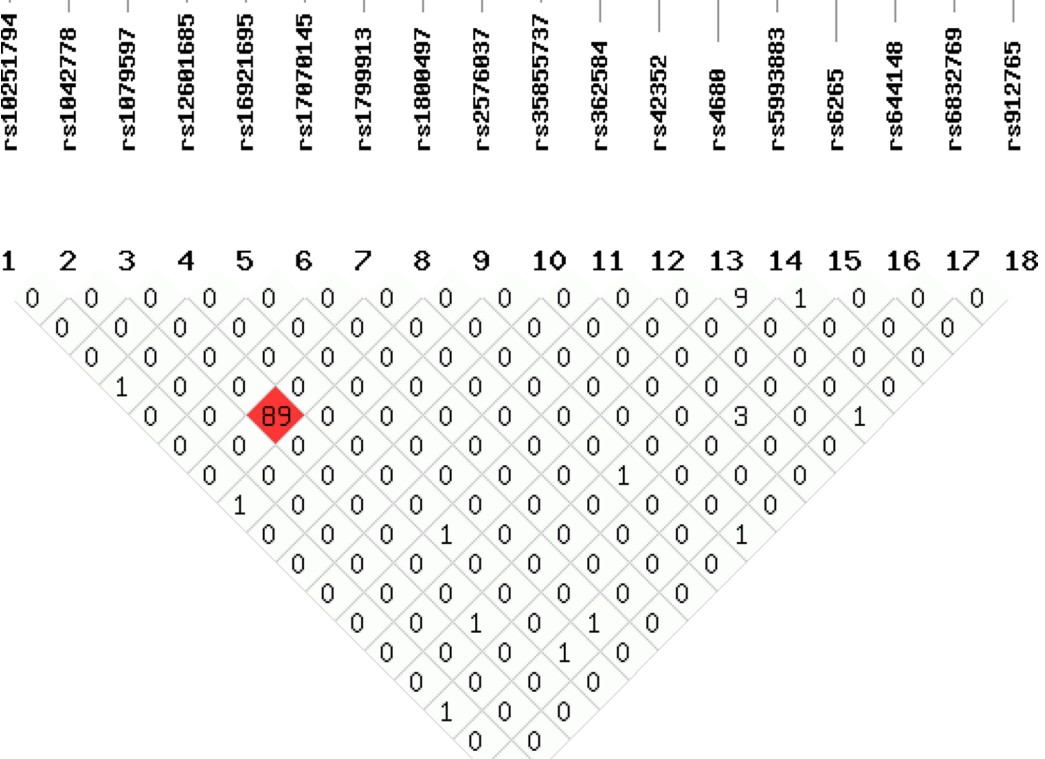

**Figure 1 Linkage disequilibrium map.** Numbers in squares indicate the degree of LD ($r^2$ value*100) between any two SNPs.

$p = 0.008$, $\eta^2 = 0.041$). One SNP (rs6832769) approached statistical significance: $F(3, 277) = 2.492$, $p = 0.06$, $\eta^2 = 0.026$. All $p$-values were uncorrected.

For each individual component score, there are significant differences between rs2576037 groups on fluency ($F = 9.731$, $p = 0.002$, $\eta^2 = 0.033$) and originality ($F = 10.996$, $p = 0.001$, $\eta^2 = 0.037$). The results remained after controlling for covariates. Post-hoc $t$-tests revealed that TT carriers scored higher on both components ($p = 0.006$ and $p = 0.002$ for fluency and originality component scores, respectively). The ANOVA and ANCOVA results are listed in Table 5.

## Genetic influences on convergent thinking

An ANOVA analysis was conducted using the RAT score as the dependent variable to identify genes correlated with convergent thinking capability. ANOVA results showed allelic variations of two SNPs significantly associated with the RAT performance: rs362584 ($F = 7.643$, $p = 0.006$, $\eta^2 = 0.026$) and rs5993883 ($F = 6.652$, $p = 0.01$, $\eta^2 = 0.023$). Post-hoc $t$-tests revealed that rs362584 GG homozygotes had lower RAT score ($p = 0.031$), so did rs5993883 TT homozygotes ($p = 0.024$). These results remained after controlling for age and intelligence as covariates, as shown in Table 6.

**Table 5 ANOVA and ANCOVA results on UUT component scores.**

| Variable | Gene | ANOVA | | | ANCOVA | | |
|---|---|---|---|---|---|---|---|
| | | $F$ | $p$ | $\eta^2$ | $F$ | $p$ | $\eta^2$ |
| Flexibility | rs10251794 | 1.373 | 0.242 | 0.005 | 0.806 | 0.37 | 0.003 |
| | rs1042778 | 3.917 | 0.049 | 0.014 | 2.674 | 0.103 | 0.009 |
| | rs1079597 | 0.003 | 0.955 | 0 | 0.038 | 0.846 | 0 |
| | rs12601685 | 0.002 | 0.966 | 0 | 0.089 | 0.765 | 0 |
| | rs16921695 | 0.358 | 0.55 | 0.001 | 0.665 | 0.416 | 0.002 |
| | rs17070145 | 0.023 | 0.879 | 0 | 0.007 | 0.935 | 0 |
| | rs1799913 | 0.152 | 0.697 | 0.001 | 0.002 | 0.963 | 0 |
| | rs1800497 | 0.219 | 0.64 | 0.001 | 0.456 | 0.5 | 0.002 |
| | rs2576037 | 0.865 | 0.353 | 0.003 | 0.531 | 0.467 | 0.002 |
| | rs362584 | 0.001 | 0.973 | 0 | 0.164 | 0.686 | 0.001 |
| | rs42352 | 0.017 | 0.896 | 0 | 0.108 | 0.743 | 0 |
| | rs4680 | 0.045 | 0.832 | 0 | 0.019 | 0.891 | 0 |
| | rs5993883 | 0.185 | 0.667 | 0.001 | 0.328 | 0.567 | 0.001 |
| | rs6265 | 0.155 | 0.694 | 0.001 | 0.431 | 0.512 | 0.002 |
| | rs6832769 | 0.002 | 0.968 | 0 | 0.017 | 0.897 | 0 |
| | rs912765 | 0.016 | 0.898 | 0 | 0.072 | 0.789 | 0 |
| Fluency | rs10251794 | 0.327 | 0.568 | 0.001 | 0.075 | 0.784 | 0 |
| | rs1042778 | 0.104 | 0.747 | 0 | 0.174 | 0.676 | 0.001 |
| | rs1079597 | 0.208 | 0.649 | 0.001 | 0.227 | 0.634 | 0.001 |
| | rs12601685 | 0.001 | 0.976 | 0 | 0.066 | 0.798 | 0 |
| | rs16921695 | 0.001 | 0.976 | 0 | 0.277 | 0.599 | 0.001 |
| | rs17070145 | 1.166 | 0.281 | 0.004 | 0.953 | 0.33 | 0.003 |
| | rs1799913 | 1.002 | 0.318 | 0.004 | 0.424 | 0.516 | 0.002 |
| | rs1800497 | 0.26 | 0.611 | 0.001 | 0.292 | 0.589 | 0.001 |
| | rs2576037 | **9.731** | **0.002** | **0.033** | **8.475** | **0.004** | **0.029** |
| | rs362584 | 0.035 | 0.851 | 0 | 0.15 | 0.699 | 0.001 |
| | rs42352 | 1.416 | 0.235 | 0.005 | 2.674 | 0.103 | 0.009 |
| | rs4680 | 0.068 | 0.795 | 0 | 0.029 | 0.866 | 0 |
| | rs5993883 | 0.384 | 0.536 | 0.001 | 0.438 | 0.509 | 0.002 |
| | rs6265 | 1.786 | 0.182 | 0.006 | 2.518 | 0.114 | 0.009 |
| | rs6832769 | 4.605 | 0.033 | 0.016 | 3.928 | 0.048 | 0.014 |
| | rs912765 | 0.004 | 0.95 | 0 | 0.119 | 0.73 | 0 |
| Originality | rs10251794 | 0.164 | 0.686 | 0.001 | 0.005 | 0.945 | 0 |
| | rs1042778 | 0.214 | 0.644 | 0.001 | 0.174 | 0.677 | 0.001 |
| | rs1079597 | 0.221 | 0.639 | 0.001 | 0.31 | 0.578 | 0.001 |
| | rs12601685 | 0.387 | 0.534 | 0.001 | 0.105 | 0.746 | 0 |
| | rs16921695 | 0.248 | 0.619 | 0.001 | 0.008 | 0.93 | 0 |
| | rs17070145 | 0.439 | 0.508 | 0.002 | 0.257 | 0.613 | 0.001 |
| | rs1799913 | 1.836 | 0.176 | 0.006 | 1.04 | 0.309 | 0.004 |
| | rs1800497 | 0.543 | 0.462 | 0.002 | 0.754 | 0.386 | 0.003 |

*(Continued)*

| Variable | Gene | ANOVA | | | ANCOVA | | |
|---|---|---|---|---|---|---|---|
| | | $F$ | $p$ | $\eta^2$ | $F$ | $p$ | $\eta^2$ |
| | rs2576037 | **10.996** | **0.001** | **0.037** | **10.056** | **0.002** | **0.035** |
| | rs362584 | 0.113 | 0.737 | 0 | 0.036 | 0.85 | 0 |
| | rs42352 | 0.967 | 0.326 | 0.003 | 2.773 | 0.097 | 0.01 |
| | rs4680 | 0.428 | 0.513 | 0.002 | 0.466 | 0.496 | 0.002 |
| | rs5993883 | 0.013 | 0.909 | 0 | 0.013 | 0.909 | 0 |
| | rs6265 | 0.516 | 0.473 | 0.002 | 0.82 | 0.366 | 0.003 |
| | rs6832769 | 4.529 | 0.034 | 0.016 | 4.496 | 0.035 | 0.016 |
| | rs912765 | 0.432 | 0.511 | 0.002 | 0.287 | 0.593 | 0.001 |

Notes:
Three component scores of UUT were analyzed by ANOVA and ANCOVA. Intelligence, age, gender and university source were used as covariables in ANCOVA. Bolded values indicate significant effect.

**Table 6 ANOVA and ANCOVA results on RAT scores.**

| Gene | ANOVA | | | ANCOVA | | |
|---|---|---|---|---|---|---|
| | $F$ | $p$ | $\eta^2$ | $F$ | $p$ | $\eta^2$ |
| rs10251794 | 2.256 | 0.134 | 0.008 | 0.918 | 0.339 | 0.003 |
| rs1042778 | 0.028 | 0.868 | 0 | 0.103[b] | 0.749 | 0 |
| rs1079597 | 0.585 | 0.445 | 0.002 | 0.463 | 0.497 | 0.002 |
| rs12601685 | 0.822 | 0.365 | 0.003 | 1.826 | 0.178 | 0.007 |
| rs16921695 | 0 | 0.996 | 0 | 0.558 | 0.456 | 0.002 |
| rs17070145 | 1.827 | 0.178 | 0.006 | 1.39 | 0.239 | 0.005 |
| rs1799913 | 1.772 | 0.184 | 0.006 | 3.773 | 0.053 | 0.013 |
| rs1800497 | 0.884 | 0.348 | 0.003 | 0.673 | 0.413 | 0.002 |
| rs2576037 | 0.084 | 0.772 | 0 | 0.001 | 0.978 | 0 |
| rs362584 | **7.643** | **0.006** | **0.026** | **5.619** | **0.018** | **0.02** |
| rs42352 | 3.395 | 0.066 | 0.012 | 1.435 | 0.232 | 0.005 |
| rs4680 | 0.369 | 0.544 | 0.001 | 0.392 | 0.532 | 0.001 |
| rs5993883 | **6.652** | **0.01** | **0.023** | **7.517** | **0.007** | **0.026** |
| rs6265 | 0.266 | 0.606 | 0.001 | 0.545 | 0.461 | 0.002 |
| rs6832769 | 0.079 | 0.779 | 0 | 0.051 | 0.821 | 0 |
| rs912765 | 0.106 | 0.745 | 0 | 0.04 | 0.841 | 0 |

Notes:
RAT scores were analyzed by ANOVA and ANCOVA. Age, intelligence, gender and university source were used as covariables in ANCOVA. Bolded values indicate significant effect.

## DISCUSSION

The present study, for the first time, investigated the genetic basis for both the divergent and the convergent thinking components of creativity. The SNPs of rs5993883 in COMT, rs362584 in SNAP25, rs2576037 in KATNAL2 were found to be associated with one of the creativity test scores. As the selection of the SNPs was based on a broader scope considering the creativity-related cognitive functions and capabilities, our results provide new information for the genetic influences on creativity.
Previous studies on the genetic basis of creativity have mainly focused on the biological mechanisms of divergent thinking, consequently limiting the exploration of possible candidate genes. By broadening the scope to include cognition-related genes, our result revealed the association between rs2576037 in KATNAL2 gene and the UUT performances on the fluency and the originality components. According to the TiGRE database, the expression level of KATNAL2 on the brain is relatively high (*Liu et al., 2008*). The production of this gene is a protein similar to the sub-unit A of the p60 katanin protein, to which axonal growth is sensitive (*Karabay et al., 2004*). Alternatively, rs2576037 might affect creativity through the medium effect of its relating conscientiousness personality (*De Moor et al., 2012*), as one recent study has reported a connection between conscientiousness and everyday creativity in a cohort of Chinese undergraduate students (*Chen, 2016*). However, none of the four SNPs reported in previous studies (rs1800497, rs1042778, rs5993883, and rs4680) showed significant influences on divergent thinking scores. This discrepancy needs to be addressed in future studies, for example, by having a more complete recording of the demographic information of the participants for a more detailed comparison between studies. Nevertheless, the present study provided evidences on additional SNPs that contribute to divergent thinking.

Our exploration of convergent thinking capabilities revealed the contributions from two SNPs, rs5993883 in COMT, and rs362584 in SNAP25. While there were no previous genetic studies directly investigating convergent thinking, the finding of rs5993883 in COMT is in accordance with one previous study on insight problem solving (*Jiang, Shang & Su, 2015*). In contrast to the insight problem tasks, RAT was used in the present study, which is believed to be a more direct measurement of convergent thinking. The rs362584 in SNAP25, however, has not been reported in previous creativity-related studies. The product of SNAP25, a presynaptic plasma membrane protein, is a key protein for the docking and fusion of synaptic and other vesicles (*Veit, Söllner & Rothman, 1996*). It is suggested that high levels of expression of this gene in specific areas of the adult brain are related to nerve terminal plasticity (*Oyler et al., 1989*; *Osen-Sand et al., 1993*). Furthermore, the genetic variants in SNAP25 have been found associated with cognitive ability (*Gosso et al., 2006*) and cognitive disorder (*Liu et al., 2017*). Since creativity is a complex cognitive process, these discoveries provided hints for the relation between rs362584 and creativity. Moreover, rs362584 has been reported to be related to neuroticism, which was proposed to be negatively related to creativity as well (*Furnham et al., 2009*).

Most importantly, we found distinct SNPs for divergent and convergent thinking, suggesting possible different genetic mechanisms behind the two processes. Such a finding is in accordance with a variety of psychological and physiological studies. For instance, the different influences of divergent and convergent thinking by working memory load argued for distinct cognitive processes underlying these two creative thinking processes (*Lin & Lien, 2013*; *Lee & Therriault, 2013*); the two creative thinking processes were associated with different personalities as well, with openness and agreeableness for divergent thinking and agreeableness for convergent thinking (*Myszkowski et al., 2015*);
moreover, the influence of chronotype and its interaction with asynchrony on convergent thinking but not divergent thinking performances, suggests potential different physiological bases (*Simor & Polner, 2017*). Therefore, our finding could be viewed as complementary to previous studies, providing the first piece of evidence toward distinct genetic mechanisms for the two creative thinking processes.

There are several limitations of the present study. First, the sample size is relatively small. The limited sample size have restricted the number of variables to be simultaneously explored, for example, the interaction between different genes, the influence of personality on genes, etc. Second, all participants were Chinese students, therefore the findings need to be interpreted with caution, especially when considering people of different races. Third, the 19 SNPs were selected considering their potential contribution to creativity-related cognitive functions and capabilities. A more comprehensive understanding of the genetic basis of creativity would require GWAS with possibly a much larger population.

## CONCLUSIONS

Taking a cognition-based perspective, the present study investigated the genetic basis for both the divergent and the convergent thinking components of creativity. Three SNPs (rs5993883, rs362584, and rs2576037) out of the selected 19 SNPs were found to be associated with creativity. Divergent and convergent thinking capabilities were related to distinct contributing SNPs. Our results provide new evidence for the genetic basis of creativity and reveal the important role of gene polymorphisms in divergent and convergent thinking.

## APPENDIX

### Unusual using test

The UUT was developed by Guilford to access examinees' divergent thinking capability (*Guilford, 1967*). Examinees were asked to list as many possible uses for common prompts as they can. In the present study, "newspaper" and "plastic bottle" were used as prompts and participants were given 5 min for each prompt.

Four independent coders (two for each prompt) blind to the identity of participants were invited to code the answers together, with discrepancies resolved by consensus. For each prompt, two coders simplified the answers by cutting unnecessary particles. For example, "be used to swat flies" and "we use newspapers to swat flies" are both coded as "swatting flies". Impossible uses and incomprehensible expressions were excluded while performing the coding. Another two coders (one for each prompt) were then invited to categorize the coded answers according to a predetermined catalog (*Qun, 2015*). Following Dippo's scoring procedure (*Dippo & Kudrowitz, 2013*): Originality score was defined as the number of uses that occur in less than 10% of all the answers; Fluency score was calculated as the total number of uses in one participant's answer; Flexibility was calculated as the total number of categories. The final score is obtained by calculating the sum of the z-score of the two prompts, which composes three components: fluency, flexibility, and originality.

### Remote associates test

Remote associates test is a well-established convergent thinking test where participants are required to find the only solution for the association among three presented cue words (*Lee & Therriault, 2013*). Here we used the Chinese version of the RAT, developed by Beijing Key Laboratory of Behavior and Mental Health (*Xiao, Yao & Qiu, 2016*). Participants were required to come up with the word associated with three presented words that appeared to be semantically unrelated. The Chinese version of RAT provided standard answers for reference. The score of RAT was defined as the number of items where a participant reached the single, correct answer. In the present study, a time limit of 5 min was given to all participants for 15 items.

### Raven's advanced progressive matrices

The APM (*Raven, Raven & Court, 1998*) is a standardized test that can be conducted either individually or in groups. It is composed of a series of matrices with the lower right corner missing. Participants are required to choose the correct answer from eight alternatives provided below to fill the matric. The complete APM includes two sets. The difficulty of 12 problems constituting Set I is equivalent to problems in the Standard Progressive Matrices, whereas the difficulty of 36 problems in Set II is greater. All problems are arranged such that the difficulty increases progressively. In this study, two problems from Set I were used as a practice, and participants answered all 36 problems from Set II in 40 min to assess intellectual efficiency.

### Real-life creative achievement

Participants' real-life creative achievement was accessed through an adjusted version of CAQ (*Carson, Peterson & Higgins, 2005*). CAQ is a self-report checklist of creativity in 10 different domains. In this study, we adjusted the questionnaire by choosing nine domains and changing the description of specific achievements to fit with Chinese college students' life. The adjusted CAQ used in the current study had good reliability (Cronbach's $\alpha$ = 0.719). For criterion-related validity, the CAQ score and UUT performance have a significant positive correlation (Pearson $r$ = 0.216, $p < 0.01$), however, the correlation between CAQ and RAT score is not significant (Pearson $r$ = −0.005, $p$ = 0.932). This part of data was not used in the present study and will be used elsewhere.

### Attention check item

One additional question was embedded in the web-based scale (for demographic information and CAQ) to check the validity of the participants' answer. In this item, we explicitly asked the participants to choose "strongly disagree" on a five-point Likert Scale from 0 (strongly disagree) to 4 (strongly agree). The participants that failed in this item were excluded from data analysis.

## ACKNOWLEDGEMENTS

The authors would like to thank Dr. Jing Qian from Tsinghua University, Dr. Xiang Li, Mr. Jin Wu from China Agricultural University for recruiting the participants,

Ms. Ning Wu, Ms. Yujin Wang from Repconex Co, for their technical support during data collection and preprocessing.

### Funding

This work is supported by the National Social Science Foundation of China (17ZDA323), the MOE (Ministry of Education, China) Project of Humanities and Social Sciences (17YJA190017), the National Science Foundation of China (U1736220), and the Tsinghua University School of Social Sciences & Institute for Data Science. The funders had no role in study design, data collection and analysis, decision to publish, or preparation of the manuscript.

### Grant Disclosures

The following grant information was disclosed by the authors:
National Social Science Foundation of China: 17ZDA323.
MOE (Ministry of Education China) Project of Humanities and Social Sciences: 17YJA190017.
National Science Foundation of China: U1736220.
Tsinghua University School of Social Sciences & Institute for Data Science.

### Competing Interests

Guihua Gong is an employee of Repconex Bio-Tech Co., Ltd.

### Author Contributions

- Wei Han conceived and designed the experiments, performed the experiments, analyzed the data, prepared figures and/or tables.
- Mi Zhang conceived and designed the experiments, performed the experiments, analyzed the data, prepared figures and/or tables.
- Xue Feng conceived and designed the experiments, analyzed the data.
- Guihua Gong conceived and designed the experiments, performed the experiments, contributed reagents/materials/analysis tools.
- Kaiping Peng contributed reagents/materials/analysis tools, authored or reviewed drafts of the paper.
- Dan Zhang conceived and designed the experiments, authored or reviewed drafts of the paper, approved the final draft.

### Human Ethics

The following information was supplied relating to ethical approvals (i.e., approving body and any reference numbers):

The study was conducted in accordance with the Declaration of Helsinki and approved by the local Ethics Committee of Tsinghua University.

### Data Availability

The raw data are provided in a Supplemental File.

## Supplemental Information

Supplemental information for this article can be found online at http://dx.doi.org/10.7717/peerj.5403#supplemental-information.

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
