# Peer review of "Genetic influences on creativity: an exploration of convergent and divergent thinking"

_PeerJ, doi:10.7717/peerj.5403_

## Round 0.1 · original submission · Major Revisions

Please revise the manuscript accordingly, try to include data analysis subtitle as suggested. Also, try to expand discussion on the importance of selected SNPs to elucidate the range of covering.

Two reviewers mentioned the gender difference. Try to (1) separate the data in present study ; and (2) include other published results for discussion.

Additionally please respond i detail to all the points by all three reviewers.

Reviewer 1 ·

Basic reporting

The present study showed a new evidence for the genetic basis on both the divergent and the convergent thinking of creativity. Moreover, the different genetic influences were found for the two components, providing possible different genetic mechanisms behind the two processes. The results are interesting and worth to be published. However, I have some concerns about the methodological and descriptive issues. Detailed comments are listed below.
Please explain in the introduction (and also the discussion) the distinction and relationship between the divergent and the convergent thinking.

Experimental design

1) The information about data analysis was missing. A paragraph of" Data analysis " is needed to explain the methods and factors used in the statistics.
2) Considering that the raw data in the creativity tests was subjective coded, the way and criterion (e.g., interrater reliability) in data coding need to be clarified.

Validity of the findings

no comment

Additional comments

1) The gender difference has been found in creativity studies. Why not take the gender factor as a covariate?
2) Only the divergent thinking were related with the real-life creative achievement. How do you explain the absence of correlation between the convergent thinking and the CAQ?
3) It seems that personality traits were considered in the selection of SNPs. Have you tried to analyze the role of personality in the genetic influences on creativity?
4) It must be "Table 2 Related cognitive functions or capabilities of selected SNPs" rather than "Table 1"

Reviewer 2 ·

Basic reporting

no comment

Experimental design

no comment

Validity of the findings

no comment

Additional comments

The present study has investigated the genetic basis of divergent and convergent thinking. The results imply the role of gene polymorphisms in the creativity. The article is well-organized and written clearly.
Several issues are listed below:
1. In Introduction, the reason why previous research mainly focused on diverge thinking could also be added.
2. The participants are chosen from two universities. Please conduct chi-square tests to avoid any confounding here, or include this variable (participant source) in the MANOVA test. Moreover, did you check the prerequisite of MANOVA? Please provide the method of correction if the prerequisite is violated.
3. In the discussion part, please relate current results with earlier findings of previous “diverge thinking” studies.
4.In Figure 1, by placing the name of polymorphisms on the left side, it is easier to understand for the reader.

Reviewer 3 ·

Basic reporting

The manuscript studied the genetic basis for divergent and convergent thinking components of creativity. Relevant literatures have been sufficiently cited and introduced. Based on the literature review, the authors proposed a new approach by taking a cognition-based perspective. Compared to previous studies mostly focusing on one single cognitive functions, here 19 SNPs were selected by incorporating a larger variety of creativity related cognitions such as memory, intelligence, personality etc. Three SNPs out of the selected 19 SNPs were found to be associated with creativity. Overall, the manuscript is well written, the findings are novel and could be of general interest in the field of psychology.

Experimental design

In my understanding, the present research is within the aim and scope of PeerJ. The research question is well defined and the design and methods were properly chosen to answer the proposed question.

While the study design is quite simple and straightforward, more explanations on the following issues are necessary:
1. The studied 19 SNPs are selected based on the introduced studies in Introduction. What was the criterion for the SNP selection? Did these SNPs provide a complete overview of possible SNPs?
2. The linkage between RAT and convergent thinking need to be further explained, with proper references. The current statement is too arbitrary.
3. The data analysis plan (e.g. zscore, MANOVA, ANCOVA etc.) should be clearly introduced in Methods, before reporting the results.
4. While age and intelligence were controlled, gender needs to be controlled as well, as it has been reported in previous studies to be associated

Validity of the findings

The sample size is relatively small (although acceptable) in gene-related studies and the studied population only covered college students, this issue should be acknowledged in Discussion. For instance, there might be significant effects in the remaining non-significant SNPs, but could not be revealed given the present sample size and population coverage.

Additional comments

Given my comments above (especially the suggested revision in methods), I suggest major revision.

---

## Round 0.2 · accepted · Accept

Congratulations ! Thanks for the efforts in improving the manuscript.

# Reviewer 1 ·

Basic reporting

This is the second review of the manuscript by Zhang et al. The current version of the manuscript is much clearer and makes the authors' points cleanly and convincingly.

Experimental design

The second version of the methods section is more clearly as well as the resutls.

Validity of the findings

The results are convincingly.

Reviewer 2 ·

Basic reporting

no comment

Experimental design

no commment

Validity of the findings

no comment

Additional comments

The study aimed to address an interesting question. Now sufficient background and further explanation is provided both in the introduction and discussion part. The analyses reported in the paper are of sufficient quality and the paper is also reasonably well-written.

Reviewer 3 ·

Basic reporting

no comment

Experimental design

no comment

Validity of the findings

no comment

Additional comments

All my concerns have been properly addressed. I am satisfied with the revision. I recommend acceptance of this manuscript.